# Child and family-focused interventions for child maltreatment and domestic abuse: development of core outcome sets

Claire Powell [1], Gene Feder [2], Ruth Gilbert [1], Laura Paulauskaite [3], Eszter Szilassy,[4] Jenny Woodman [5], Emma Howarth [6]

**Correspondence to**
Dr Claire Powell;
c.powell@ucl.ac.uk

## ABSTRACT

**Background** The current evidence for child maltreatment (CM) and domestic violence and abuse (DVA) interventions is limited by the diversity of outcomes evaluated and the variety of measures used. The result is studies that are difficult to compare and lack focus on outcomes reflecting service user or provider priorities.

**Objective** To develop core outcome sets (COSs) for evaluations of child and family-focused interventions for: (1) CM and (2) DVA.

**Design** We conducted a two-stage consensus process. Stage 1: a long list of candidate outcomes across CM and DVA was developed through rapid systematic reviews of intervention studies, qualitative and grey literature; stakeholder workshops; survivor interviews. Stage 2: three-panel, three-round e-Delphi surveys for CM and DVA with consensus meetings to agree with the final COSs.

**Participants** 287 stakeholders participated in at least one stage of the process (ie, either CM or DVA COS development): workshops (n=76), two e-Delphi surveys (n=170) and consensus meetings (n=43). Stakeholders included CM and DVA survivors, practitioners, commissioners, policymakers and researchers.

**Results** Stage 1 identified 335 outcomes categorised into 9 areas and 39 domains. Following stage 2, the final five outcomes included in the CM-COS were: child emotional health and well-being; child's trusted relationships; feelings of safety; child abuse and neglect; service harms. The final five outcomes in the DVA-COS were: child emotional health and well-being; caregiver emotional health and well-being; family relationships; freedom to go about daily life; feelings of safety.

**Conclusions** We developed two COSs for CM and DVA with two common outcomes (child emotional health and well-being; feelings of safety). The COSs reflect shared priorities among service users, providers and researchers. Use of these COSs across trials and service evaluations for children and families affected by CM and DVA will make outcome selection more consistent and help harmonise research and practice.

## INTRODUCTION

There is insufficient high-quality evidence for the effectiveness of child maltreatment (CM) and domestic violence and abuse (DVA) interventions for improving child and family outcomes.[1–3] This means service providers do

## STRENGTHS AND LIMITATIONS OF THIS STUDY

⇒ This study used a systematic consensus process which involved a large number of practitioners, policymakers, researchers and adult survivors of violence and abuse.
⇒ The use of multiple forms of engagement enabled participants to take part in a range of ways: workshops, interviews, online survey.
⇒ Survivor involvement was central to the process and enabled us to shape the study as we carried it out.
⇒ Given the impact of the COVID-19 pandemic, we were unable to reach as many frontline domestic violence and abuse and child maltreatment organisations as we intended.
⇒ Our reach into minoritised communities was not as extensive as we would have liked, so survivors in the study do not reflect the extent of diverse communities in the UK. We sought explicit feedback on this limitation in the final stage of the study.

not know which interventions are most useful or might potentially cause harm: this uncertainty could discourage identification of CM or DVA.[4] It is difficult to compare and synthesise studies due to inconsistent outcome reporting and the range of measures used. Even widely reported outcomes such as depression or experience of violence are measured in varied ways,[4–6] including across evaluations of similar interventions such as psychological therapies.[7] Current global guidance on evaluation of CM programmes does not promote the use of comparable outcomes.[8]

Decisions about which outcomes to measure tend to be led by researchers, meaning those selected may not be relevant to service users and providers. For example, Howarth *et al*[1] reported that clinical trials of interventions for CM or DVA prioritised symptoms and diagnoses, which differed from priorities of affected children and families who emphasised wider outcomes related to everyday well-being and functioning.[9] Similarly, work with

service users and providers of domestic violence perpetrator programmes found broader definitions of 'success' than are traditionally measured.[10]

Inconsistent or inappropriate outcome reporting and measurement result in research wastage[11] and uncertainty about what interventions work or do not work, and for whom. Consequently, funding might be wasted on ineffective or even harmful interventions.

One way to address these challenges is development of core outcome sets (COSs), a minimum set of outcomes which are reported across all clinical trials, and potentially practice-based evaluations as well. COSs should be developed with a standardised consensus process to identify outcomes important to all stakeholders, resulting in the selection of the core outcomes.[10] The aim of a COS is to increase the consistency of outcome measurement and reporting in order that evidence accumulates in helpful ways while minimising selective outcome reporting, and ensuring the outcomes are meaningful to service users and providers.

### Aim
Our aim was to develop two COSs to be used in evaluations of child, parent or whole-family interventions for: (1) CM and (2) DVA. For DVA, we focused on interpersonal violence and abuse between parents/caregivers which is the most prevalent form of DVA.[12] By developing the two sets in parallel, we intended to bring together two areas of practice that may result in a joint service response,[13 14] although they often operate separately.[15] It is acknowledged that collaboration between researchers and practitioners in this area is vital,[16] and that the DVA and CM research agendas should be brought together,[17] not least because exposure to DVA is considered by many to be a form of CM, and in up to 60% of homes with DVA, there may also be CM present.[18]

### Scope
The COSs were developed for research and evaluation of any psychosocial interventions (as defined by the Institute of Medicine[19]) for children and families with experience of or at risk of experiencing CM or DVA. We included any intervention globally that aimed to improve child outcomes through targeting parents or family members. The target population was children aged under 19 years with current or previous experience of CM or DVA. Included interventions could be delivered in any setting, to an individual, dyad or group, and any combination of child, parent/caregiver, family groups alone or in combination. To be in scope, those eligible for the intervention had to have been exposed to/experienced CM or DVA or to be at increased risk of CM or DVA.

### METHODS
We registered the study with the Core Outcome Measures in Effectiveness Trials (COMET) initiative[20] and published the study protocol.[21]

Following COMET methodology,[22] we used a two-stage approach. In stage 1, we devised a long list of candidate outcomes from stakeholder consultation, qualitative interviews, trials and the wider literature. In stage 2, two e-Delphi consensus processes based on this long list resulted in two COSs. This report of the process and the findings follows Core Outcome Set-STAndards for Reporting (COS-STAR guidelines)[23] (see figure 1 for study flow diagram).

We decided to run stage 1 jointly for CM and DVA to develop a comprehensive long list of outcomes that could be relevant to all family-focused solutions and to reflect the high levels of co-occurrence between CM and DVA. However, given differences in current approaches to CM and DVA interventions, for example, social care interventions for CM and women's refuge interventions for DVA, we ran stage 2 as two separate e-Delphi processes to understand differing priorities between these two fields.

### Patient and public involvement
We consulted two survivor advisory groups, one comprising adult survivors of DVA, the other comprising young adult care leavers (for details, see the Acknowledgements section). We involved both groups after their participation in the stakeholder workshops and they provided advice on the accessibility and appropriateness of study materials, for example, survey wording. They also designed a sensitivity protocol for the qualitative interviews to ensure that these were trauma informed, consulted on the facilitation of and participated in the consensus workshops (see online supplemental material 1 for sensitivity protocol).

### Participants
We recruited from the following groups of participants for the stakeholder workshops (stage 1), qualitative interviews (stage 1) and for the e-Delphi survey and consensus workshops (stage 2):

a. Survivors: UK-based adults with lived experience of CM/DVA in childhood or as the parent of a child who experienced CM/DVA. Survivors did not have to have used any services or interventions to take part, and were recruited through survivor networks, charities and university patient and public involvement (PPI) groups.

b. Practitioners: UK-based professionals working in frontline (ie, delivering services) or second-tier (ie, supporting service providers) CM/DVA organisations, local authority commissioning or working in any policy capacity in this area. We recruited through research team networks, directly approaching organisations and social media.

c. Researchers: English-speaking academic researchers based in high-income country universities or independent research organisations in the UK and internationally. We recruited through research networks, directly approaching researchers or teams, and social media

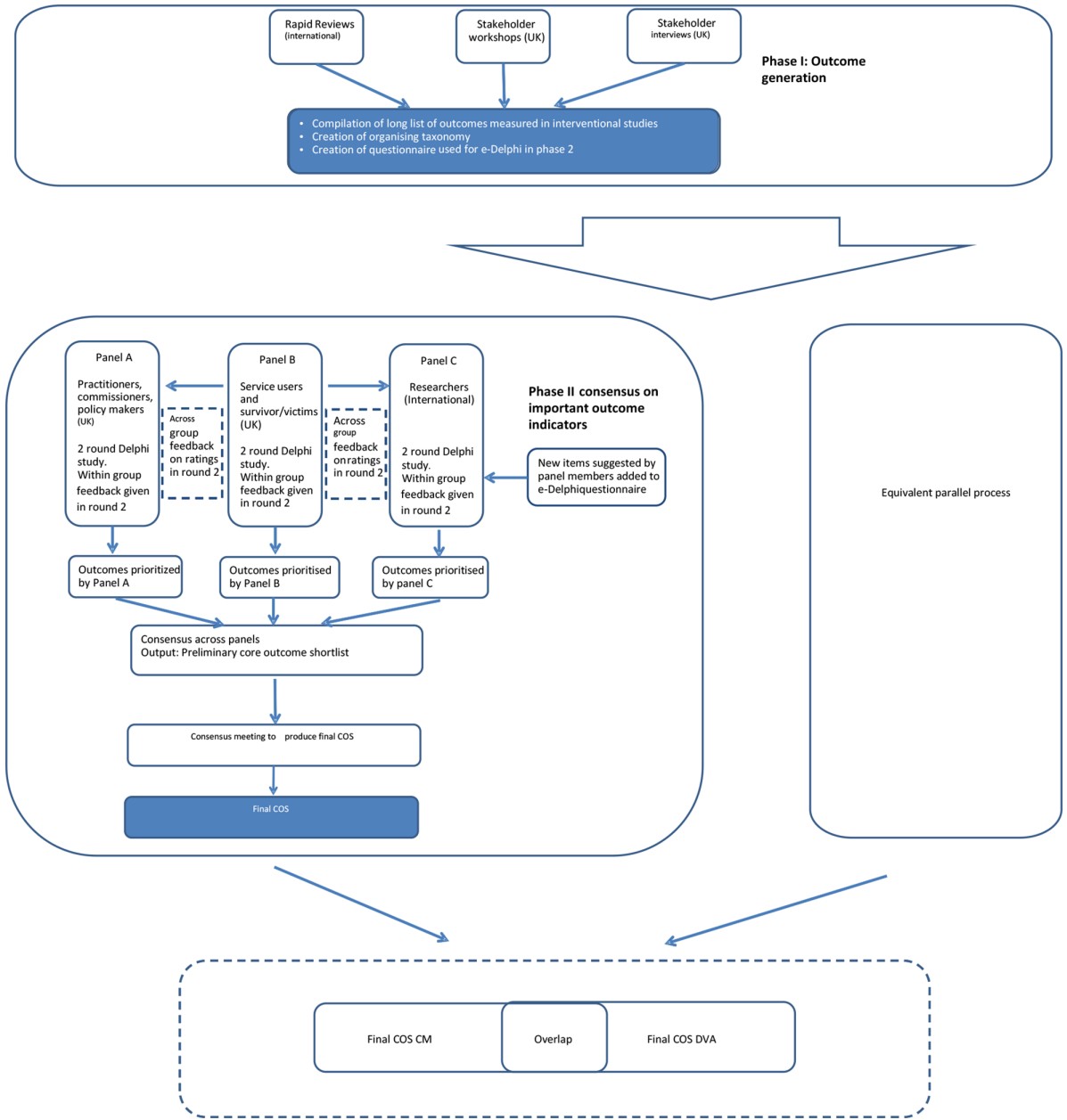

**Figure 1** Study design flow chart. CM, child maltreatment; COS, core outcome set; DVA, domestic violence and abuse.

(see online supplemental material 2 for stage 1 and stage 2 participant details).

### Stage 1: information sources

In stage 1, we identified candidate outcomes from three sources: (1) rapid evidence reviews, (2) consultation with key stakeholders and (3) qualitative interviews.

#### Rapid evidence reviews

We carried out a series of rapid reviews separately for CM and DVA, updating previous systematic reviews[1 2] (see online supplemental material 3 for inclusion criteria and online supplemental material 4 for review flow charts and for the search strategies, see protocol[21]). For the intervention literature, we searched Medline, Embase, PsycInfo, Cochrane and Web of Science from January 2014 to May 2019. For the qualitative literature, ASSIA, CINAHL,

GoogleScholar, PsycInfo and SSCI were searched from October 2015 (DVA) and July 2014 (CM) until October 2019. Differences in dates reflect differences in original reviews being updated. The grey literature review was carried out for CM and DVA simultaneously, to identify additional outcomes. We searched websites of relevant organisations (see online supplemental material 5 for full list) and NICE Evidence Search and Open Grey databases.

Two researchers (CP and EH) dual screened 400 title/abstracts for each of the DVA and CM (200 from the intervention literature searches and 200 from the qualitative literature searches) as a consistency check. The first 10% of full texts were dual screened (CP), with disagreements resolved by discussion as a further consistency check, prior to screening of the rest of the full texts by a single researcher. Study details and outcomes were extracted

by a single researcher with 5% cross-checked by the lead researcher. Participant quotations and author-identified themes from the qualitative studies and grey literature were extracted into a matrix to identify outcomes. All outcomes identified from the reviews were added to the long list and de-duplicated as part of the process.

### Consultation with key stakeholders

We carried out two half-day workshops (19 June 2019 for DVA, 5 September 2019 for CM) in which a total of 76 participants generated a list of candidate outcomes in small groups. The research team and external speakers gave presentations in the workshops on CM and DVA intervention research and interim findings from the rapid evidence reviews. These were followed by small group discussions on definitions of key terms and outcomes. All outcomes from small group discussions were de-duplicated and added to the long list.

### Qualitative interviews

To identify additional outcomes, including those that survivors would like to be measured but are not currently, one researcher (CP) carried out 10 semistructured interviews in August–December 2020 with survivors of CM and/or DVA. The interviews were designed in consultation with the study survivor advisory groups and participants gave written, informed consent before each interview, with the option to withdraw at any stage. Interviews lasted 40–65 min, were audio-recorded and transcribed. One interviewee chose to participate by email. Two researchers (CP and LP) extracted outcomes from the interview transcripts, using the questions: (1) Is this an outcome? and (2) Is this related to an outcome? Outcomes were cross-checked with the long list and added if they were not already present.

### Stage 1: long list and taxonomy development

A single long list was produced comprising all outcomes gathered from CM and DVA information sources (ie, rapid evidence reviews, stakeholder consultations and qualitative interviews). Producing the final long list involved an iterative process of (1) de-duplicating outcomes produced within each information source (eg, the consultation workshops); (2) combining outcome lists from information sources to form a single long list; (3) grouping similar outcomes and developing categories to label groups of similar outcomes; (4) de-duplicating outcomes across the long list; (5) combining specific outcome indicators together. Exact duplicates were dropped first, and similar outcomes were reworded to reflect broader meanings. For example, the child health outcome 'sleep' was created by combining: 'amount of sleep', 'quality of sleep', 'experience of nightmares', 'sleep routine', 'insomnia' and 'sleep-walking'. This process was carried out by two researchers (CP and EH) cross-checking and refining the long list. At several points, the PPI, expert advisory groups, and wider research team provided feedback on specific sections of the list and gave input on the

level of detail for the outcomes, for example, creating one 'sleep' outcome as described above, rather than two.

We organised the long list of outcomes into a taxonomy using an iterative team-based approach. The initial categories, based on participant discussion and notes produced in the stakeholder workshops, were structured using Bronfenbrenner's ecological model as a framework.[24] Related theoretical frameworks for CM and DVA were consulted[25–27] and used to further refine the taxonomy, so it accurately reflected the long list. The taxonomy was finalised by the expert advisory group and the survivor groups reviewed the final categories and outcome wording for sense and missing outcomes. The final long list and taxonomy were used as the basis for the questionnaires in the e-Delphi survey and final consensus workshop (see figure 2 for outcome identification and long list compilation flow chart).

### Stage 2: consensus process

In stage 2, participants took part in a three-panel, three-round e-Delphi survey to reach consensus. The three panels were: survivor, researcher, practitioner (as described above). Participants were informed that the operationalisation and measurement of each indicator would take place in a subsequent, as yet unfunded study, and reflects the development process of other outcome sets. We used Qualtrics software (Qualtrics, Provo, Utah, USA) to collect the data. PPI groups reviewed and edited the survey questions and format for clarity. Survey questions were piloted with one survivor and two team members not involved in its design (LP and RG). Two parallel surveys were conducted, one for CM and one for DVA. Participants gave informed consent as part of the survey, with the option to withdraw their data up to two weeks after completion (see online supplemental material 6 for survey and figure 3 for stage 2 e-Delphi consensus process and inclusion cut-offs).

We asked survivor participants whether they were survivors of CM or DVA. If they had experienced both, they chose whether to participate in the CM or the DVA survey. We assigned professionals to the research or practitioner panels based on information these participants supplied. Although we ran the two surveys in parallel, the first round of the CM and DVA surveys was the same. However, the second round (and subsequently third round and consensus workshop) differed according to the domains and outcomes selected by panels in the CM and DVA surveys. The first round of both surveys asked participants to select broad outcome domains to reduce the long list. The second round offered the more specific outcome indicators from the selected domains, and the third round offered the remaining outcome indicators. Outcome domains and indicators were presented in a random order to participants in each round to reduce bias. At the end of each survey, we had two short lists of highly prioritised outcomes, one for CM and one for DVA.

The final stage of the consensus process was online consensus workshops with participants representing the

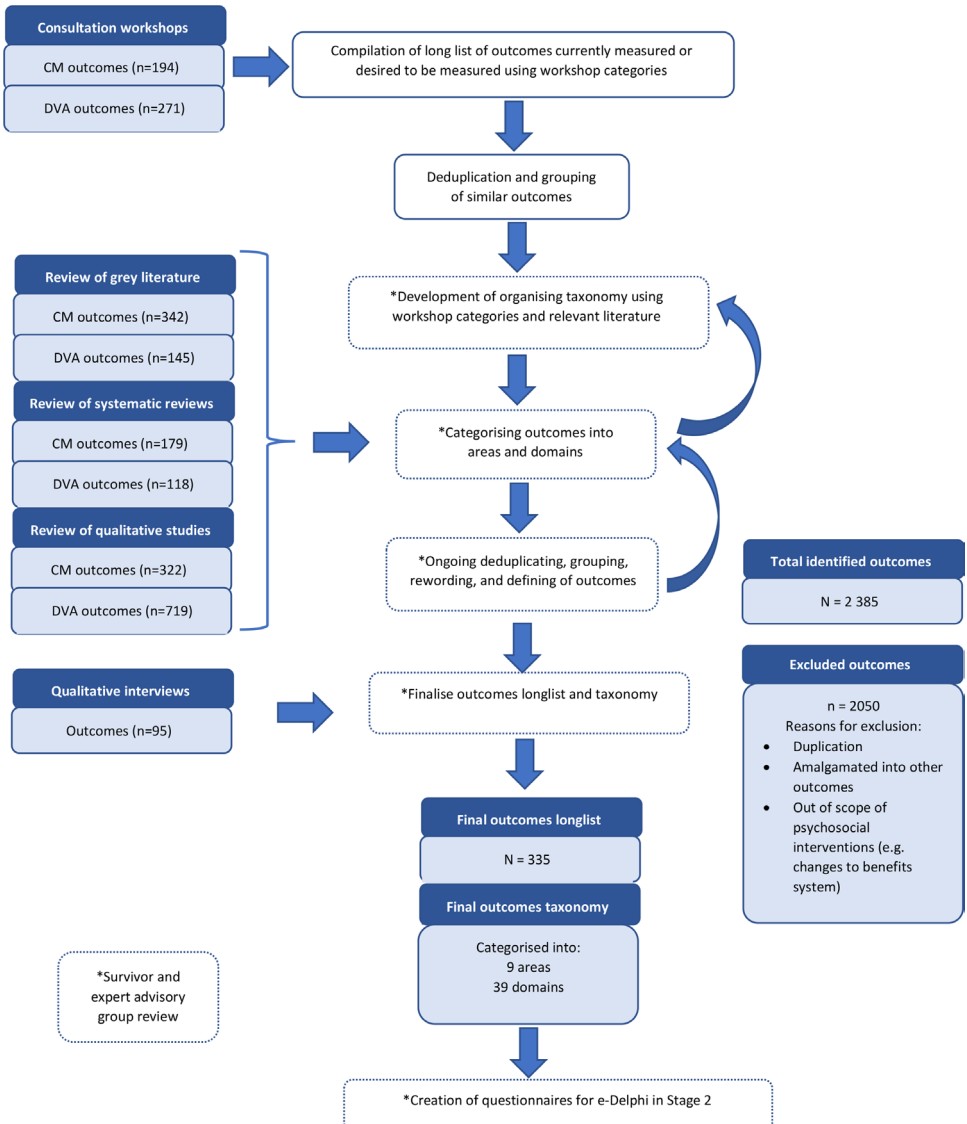

**Figure 2** Stage 1: outcome identification and long list compilation flow chart. CM, child maltreatment; DVA, domestic violence and abuse.

three panels: survivor, practitioner, researcher. COVID-19 restrictions meant the workshops could not be held in person, therefore we invited 24 participants to each workshop (ie, one for DVA and one for CM), drawing on current recommendations for online consensus development meetings from the James Lind Alliance.[28] Information on the purpose of the workshop and the relevant short lists were sent to all participants ahead of the workshop. Participants discussed their top and bottom three outcomes in two rounds of breakout group sessions, before voting on which to include.

The final COSs included only those outcomes that at least half of workshop participants voted to include. There was a final plenary discussion on the COSs and the implications for marginalised and underheard groups. To maintain survivor confidentiality, the workshops were not recorded; however, workshop facilitators made detailed notes of discussions. An external facilitator led the workshops to maintain impartiality and members of the research team

(CP, EH, ES) acted as neutral co-facilitators of breakout groups. A qualified counsellor was available to speak to survivors during the workshops and for a week afterwards. Further details about how we adapted the process to be trauma informed will be described in future work.

## RESULTS

This study was completed according to the study protocol, any changes were made in response to the evolving COVID-19 situation (eg, holding interviews and workshops online) and to reduce participant burden (eg, increasing the survey consensus threshold because the levels of consensus across outcomes were higher than expected) (for full details, see online supplemental material 7).

### Stage 1

Six rapid evidence reviews, two stakeholder workshops and 10 survivor interviews identified 335 unique outcomes

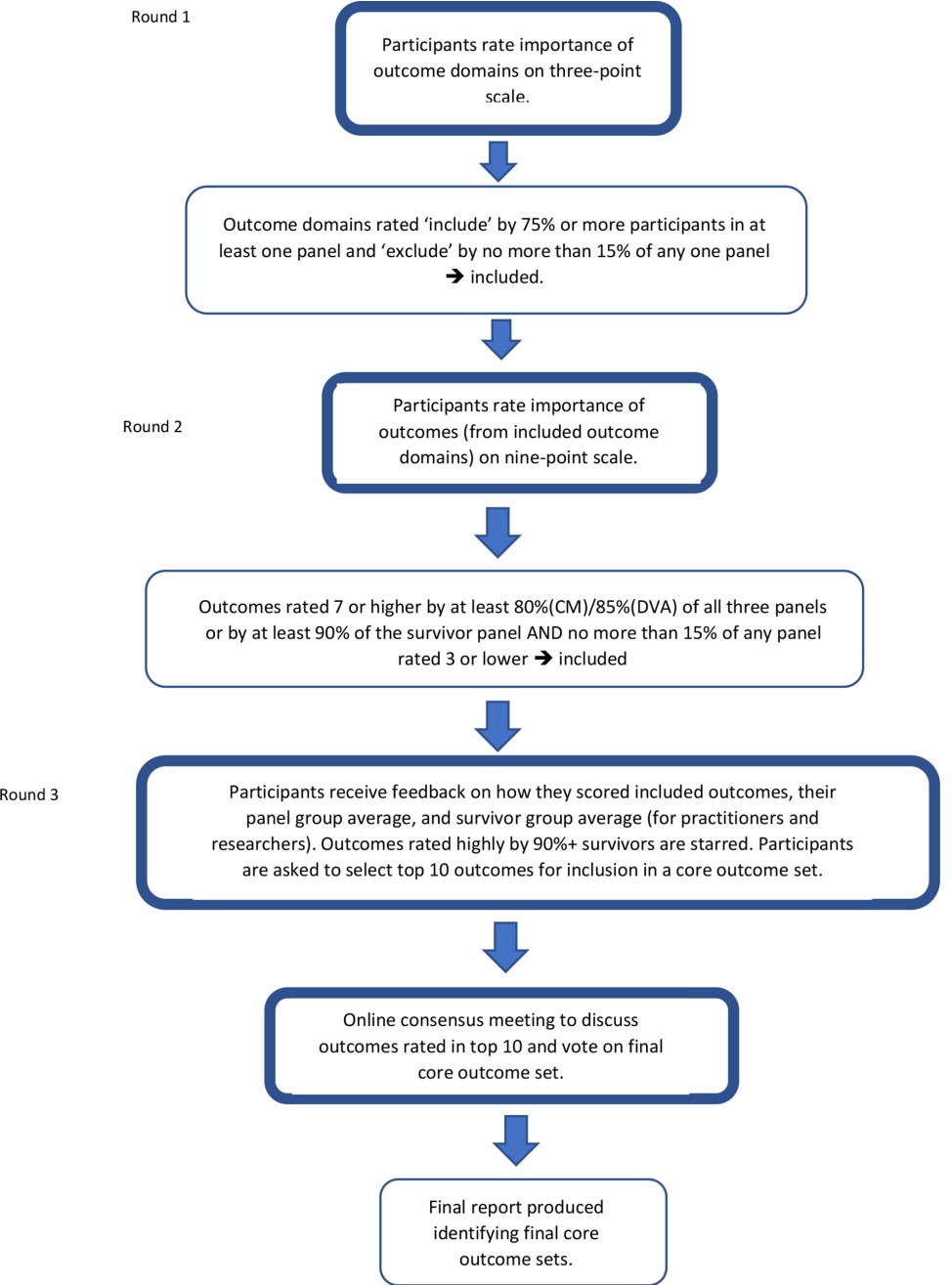

Round 1

Participants rate importance of outcome domains on three-point scale.

Outcome domains rated 'include' by 75% or more participants in at least one panel and 'exclude' by no more than 15% of any one panel ➔ included.

Round 2

Participants rate importance of outcomes (from included outcome domains) on nine-point scale.

Outcomes rated 7 or higher by at least 80%(CM)/85%(DVA) of all three panels or by at least 90% of the survivor panel AND no more than 15% of any panel rated 3 or lower ➔ included

Round 3

Participants receive feedback on how they scored included outcomes, their panel group average, and survivor group average (for practitioners and researchers). Outcomes rated highly by 90%+ survivors are starred. Participants are asked to select top 10 outcomes for inclusion in a core outcome set.

Online consensus meeting to discuss outcomes rated in top 10 and vote on final core outcome set.

Final report produced identifying final core outcome sets.

**Figure 3** Stage 2: e-Delphi consensus process to develop and prioritise core outcomes. CM, child maltreatment; DVA, domestic violence and abuse.

(for long list, see https://osf.io/yhnfq/). The outcomes taxonomy comprised 39 domains nested within 9 broad areas which were: (1) child health and well-being; (2) caregiver health and well-being; (3) caregiver relationships and parenting; (4) home environment and household; (5) social support and peer relations; (6) community resources and institutions; (7) safety, feelings and knowledge related to violence and abuse; (8) violence, abuse and maltreatment; (9) intervention outcomes (for the full taxonomy, see https://osf.io/9htz4/).

## Stage 2

We recruited a total of 80 participants for the CM e-Delphi survey and 90 participants for the DVA e-Delphi survey

(not all participants took part in all rounds). We did not reach our recruitment target of 30 participants in each panel; we think this was primarily due to the impact of COVID-19 and associated delays which resulted in the survey running over the school summer holidays (see figure 4 for the distribution of participants by round in stage 2).

Outcomes were dropped at each stage of the e-Delphi. No new outcomes were added throughout the survey process; however, suggestions offered by participants (in free text within the survey) were cross-checked with the long list and incorporated as details into pre-existing outcomes, for example, 'safety within court proceedings'

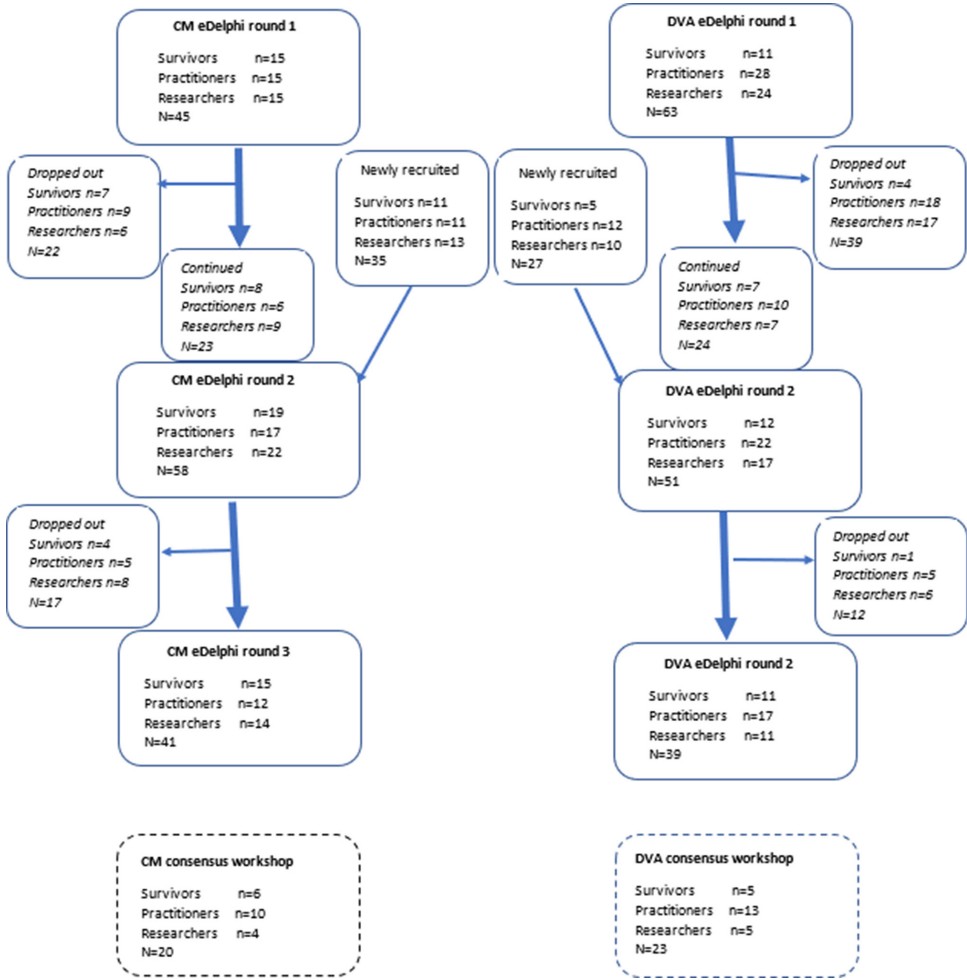

**Figure 4** Stage 2: consensus process participant flow chart. CM, child maltreatment; DVA, domestic violence and abuse.

was added to the 'safety' outcome. Two domains were merged because of feedback from the survey—'child-specific exposure to DVA' was incorporated into 'child maltreatment' (see figure 5 for the selection of outcomes and online supplemental material 8 for all ratings of survey items by panel and by round).

Given ongoing consultation throughout the process regarding practitioner and researcher concerns about the potential size of the COS, the research team agreed that the COS would be limited to five outcomes, with discussion around any tied scores. Practitioners and researchers were concerned about the possible burden if the outcomes in the final COS were very different from those already collected in the context of service monitoring. Through informal discussions with collaborators, five was agreed as enough to capture and compare shared outcomes but a small enough number to be feasible to implement in research and evaluation. In the event, both consensus workshops yielded five outcomes that scored higher than the rest and met our inclusion criteria that at least 50% of participants voted for them.

Following the final round of discussion in the DVA consensus meeting, it was agreed to change *child mental health* to *child emotional health and well-being* because participants felt this reflected both survivor and practitioner

perspectives in a more holistic way. Workshop participants agreed to a consensus statement to document this change (see online supplemental material 9).

### Core outcome sets
The final COSs, each comprising five outcomes, are as follows:

CM-COS:

1. Child emotional health and well-being: includes emotions, mood, internalising problems, emotional regulation, emotional security and emotional numbness.
2. Child's trusted relationships: includes with friends, family, other adults; network of trust adults—includes access to, quantity and quality, in and out of school.
3. Service harms: includes general harmful service response, iatrogenic harm, replication of abusive dynamics in therapy, retraumatisation, revictimisation, secondary abuse, intervention adding to self-blame in women, traumatic physical procedures.
4. Feelings of safety: for non-abusive parent and child; global safety, including psychological, physical, body, family, neighbourhood around perpetrator, at home, at school, in the community, on social media, from abusive individuals, from child removal, from court proceedings.

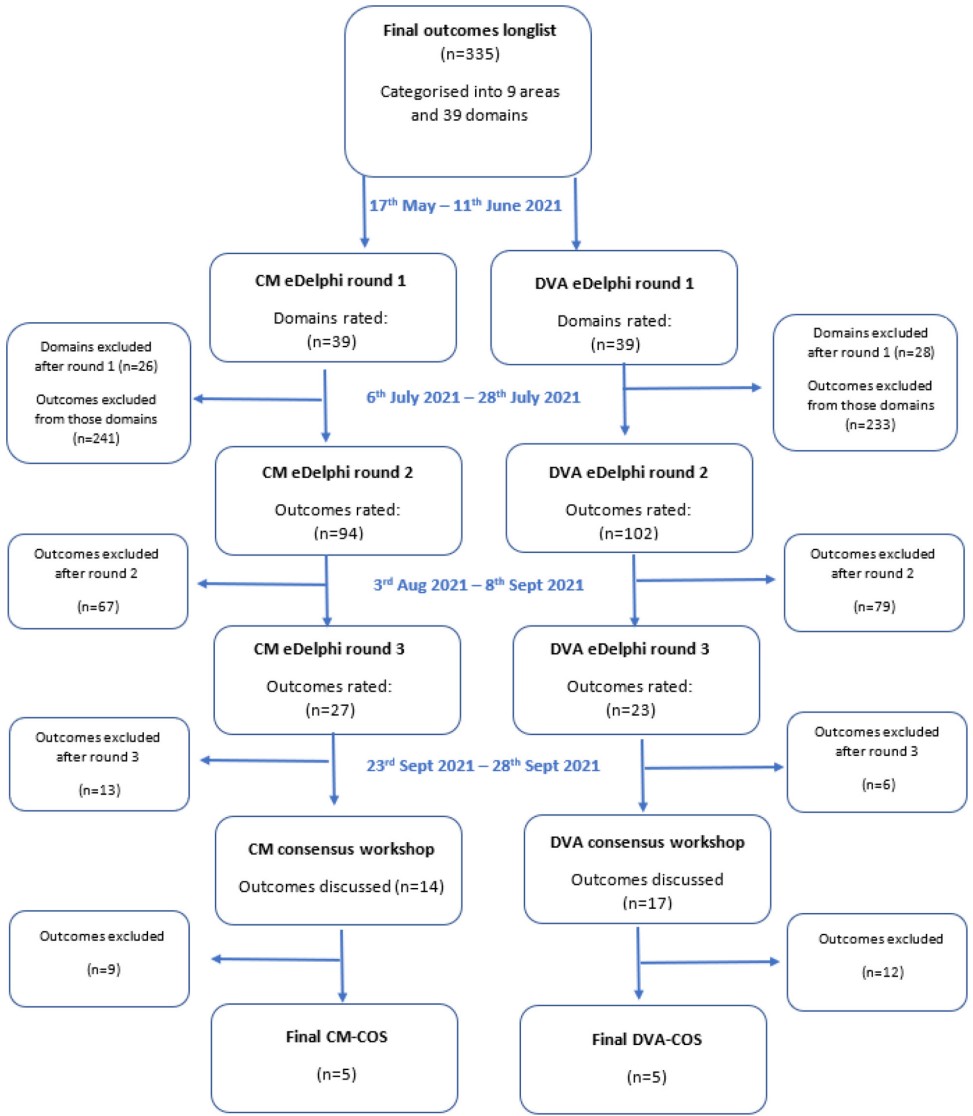

**Figure 5** Stage 2: selection of outcomes flow chart. CM, child maltreatment; COS, core outcome set; DVA, domestic violence and abuse.

5. Child abuse and neglect: includes occurrence, recurrence, risk, type.

DVA-COS:

1. Child emotional health and well-being: includes emotions, mood, internalising problems, emotional regulation, emotional security and emotional numbness.
2. Feelings of safety: for non-abusive parent and child; global safety, including psychological, physical, body, family, neighbourhood around perpetrator, at home, at school, in the community, on social media, from abusive individuals, from child removal, from court proceedings.
3. Caregiver emotional health and well-being: includes emotional functioning, emotional reactions, emotions, emotional self-regulation, control over emotions, ability to connect to emotions, mood, frame of mind, general sense of well-being.
4. Family relationships: includes overall family relationships and functioning, quality and type of relationships, feeling closer as a family, family conflict resolu-

tion, feeling closer to children, changes after leaving abusive partner; sibling relationships including after separation; child relationship with birth and foster/adoptive families.
5. Freedom to go about daily life: includes ability to get home safely from school/work/friends/family, etc (see figure 6 to see the COSs as an infographic).

The outcomes are ordered by number of workshop votes (for ranking of all discussed outcomes, see online supplemental material 10).

### COSs and underserved groups

Following agreement on the final sets, the afternoon discussion focused on who might be excluded by the COSs. The key groups included: neurodiverse and disabled children and families, families living in poverty, and ethnic and racialised minorities. Discussion centred on three topics: (1) language and meaning; (2) rights and discrimination; (3) practical delivery of relevant

## Child maltreatment

- **Child abuse and neglect**
  includes all harms caused to a child by adults in a position of responsibility

- **Service harms**
  Any retraumatisating effects of the intervention

- **Child's trusted relationships**
  Positive relationships a child has with any adults

- **Child emotional health & wellbeing**
  All aspects of emotional and mental health

- **Feelings of safety**
  Includes psychological and physical safety

- **Freedom to go about daily life**
  For example, getting home safely from school

- **Family relationships**
  Quality and type of relationship with birth/foster/adoptive family

- **Caregiver emotional health & wellbeing**
  All aspects of emotional and mental health

## Domestic violence

ucl.ac.uk/children-policy-research

**Figure 6** Final core outcome sets. NIHR, National Institute for Health Research.

measurement tools (for further details, see online supplemental material 11).

## DISCUSSION

We developed two COSs using consensus methods for child and family-focused interventions for CM and DVA. The scope of the outcome sets was broad, including health and well-being, safety and relationship outcomes. The COSs had two outcomes in common—*child emotional health and well-being* and *feelings of safety*—reflecting shared priorities between service users and providers across CM and DVA. Furthermore, three of the eight core outcomes are not currently reported in the CM or DVA intervention literature: *child's trusted relationships, freedom to go about daily*

*life* or *service harms.* This emphasises the importance of a consensus process involving service users and providers, to capture outcomes critical to all stakeholders, not just researchers.

This is the first time that COSs have been developed for CM and DVA, reflecting a methodological leap forward for the fields. The overlapping nature of the COSs reflects growing recognition that these experiences co-occur within families and that service and therapeutic responses need to consider both CM and DVA. This is novel given research literature and service provision have largely developed in parallel.

The next step is to develop consensus and guidance about how best to further operationalise and measure outcomes consistent with the domains we identify, three of which have not before been included in any quantitative research or service evaluations. In the longer term, consensus around the best measures to use to capture outcomes will facilitate meta-analyses of outcomes which require the same measure to have been used across studies. In the shorter term, our COSs can be used as a framework for service evaluation and research. Even if different measures are used across service evaluations, narrative methods can be used to synthesise findings across studies within a core outcome. This represents a significant step forward.

Additionally, using our COSs as a guide to both developing and evaluating interventions will mean that interventions in this field will have a better chance of making a meaningful difference to the lives of those experiencing DVA or CM given they are aligned with the priorities of representatives of these groups. This two-stage process of determining 'what to measure' followed by 'how' reflects the development process of many other COSs and is dictated by pragmatic considerations such as securing funding to undertake both parts and also the desire to communicate useful interim findings to stakeholders.

The key strength of our COS development process was the scope of engagement: we involved survivors at every stage both as participants and in an advisory capacity, and we involved professionals from a range of organisations across the non-governmental and statutory sectors. As a result, we identified a broad range of possible outcomes, including those that are not currently measured but nonetheless considered important. The high level of consensus about the importance of many outcomes was unexpected (and resulted in changes to the protocol), highlighting that there are many shared priorities across survivors, service providers and researchers. The outcomes identified in this study are consistent with findings from engagement with a range of stakeholders for DVA perpetrator programmes.[10]

By developing COS in parallel for CM and DVA, and finding important areas of overlap, such as safety, our findings will help underpin evaluations of family-focused interventions. The shared priorities across the COSs highlight the joint importance of health and well-being outcomes rather than a focus on mental health and

psychopathology, as well as relationships and safety. The stakeholder involvement and transparent development process have resulted in cross-sector engagement which will be useful for future implementation. Without extensive survivor engagement and reviewing literature beyond clinical studies, the COSs would not have reflected all stakeholder priorities.

## Limitations

We faced challenges at every stage because of the COVID-19 pandemic and because we were adapting a process developed to support the evaluation of clearly defined medical and surgical interventions, rather than complex psychosocial interventions. COVID-19 restrictions meant we were unable to access as many frontline services as we intended, and it likely affected the survey drop-out rates. Moving online meant we only included survivors who could access the internet, so we did not reach the most marginalised groups of survivors. This limitation needs to be addressed in any future development of CM-COS and DVA-COS.

For these COSs to be used in applied research, the concepts need further refinement and operationalisation via an explicit consensus process (with survivors, practitioners and researchers) to identify appropriate measurement tools.[29] The COSs as they currently stand represent the first stage of harmonising outcome measurement within and across CM and DVA trials and service evaluations. In order for the COSs to be used widely across interventions, service providers may need to change what they are measuring and how, which will potentially involve additional costs and staff training. However, UK-based family and children's services already collect outcomes to support service planning and delivery,[30–32] so our hope is the CM-COS and DVA-COS would supplement preexisting frameworks and support their harmonisation.

Future work needs to consider the international and cross-cultural relevance of the outcomes and their definition, including their application to minoritised groups and families living in varied socioeconomic situations.

## CONCLUSION

Our COSs represent an important first step in developing consistent outcome measurement strategies within and across these two related fields, and advancement towards inclusion of outcomes most important to all stakeholders. Our hope is that researchers will be able to more easily compare outcomes across studies and the evidence base can be synthesised and thus build cumulatively; the ultimate goal being that better quality evidence is available to assist decision-makers regarding which services and interventions to support.

## Author affiliations
[1]Population, Policy and Practice Research and Teaching Department, UCL Great Ormond Street Institute of Child Health, University College London, London, UK
[2]Centre for Academic Primary Care, Population Health Sciences, Bristol Medical School, Bristol University, Bristol, UK
[3]Institute of Education, University College London, London, UK
[4]Centre for Academic Primary Care, Population Health Sciences, Bristol Medical School, University of Bristol, Bristol, UK
[5]Social Research Institute, University College London, London, UK
[6]School of Psychology, University of East London, London, UK

**Acknowledgements** We wish to thank all the survivors who have taken part in every stage of the study; we are grateful to everyone who used their lived experience to improve this work. Further thanks are due to our survivor involvement groups: VOICES (a survivor-led charity for women who have experienced domestic abuse) and Voices from Care (a charity dedicated to the rights of care experienced young people) in partnership with Rachael Vaughan, Cardiff University. We are also grateful to the health and social care professionals and researchers who have participated and the ongoing advice from our steering group: David and Elisabeth Carney Haworth, Hannah Edwards, Victoria Jepson, Elaine Fulton, Dr Deborah Hodes, Professor Sally Kendall and Professor Carol Rivas. Finally, we thank our independent consensus workshop facilitator, Katherine Cowan, and our workshop counsellor, Dr Alison Gregory, for their input.

**Contributors** EH conceived of the original study design, which was refined and developed by EH, CP, RG, GF, ES and JW. CP, EH and LP designed data collection tools and collected data for the study. CP cleaned and analysed the data and drafted the paper. EH, RG, GF, ES, JW, CP and LP revised the paper. RG acted as guarantor during the study.

**Funding** This study is funded by the National Institute for Health Research (NIHR) Policy Research Programme (funder reference: PR-PRU-1217-21301; UCL award code: 177763).

**Disclaimer** The views expressed are those of the authors and not necessarily those of the NIHR or the Department of Health and Social Care.

**Competing interests** None declared.

**Patient and public involvement** Patients and/or the public were involved in the design, or conduct, or reporting, or dissemination plans of this research. Refer to the Methods section for further details.

**Patient consent for publication** Not required.

**Ethics approval** Ethics approval was provided by University College London's Research Ethics Committee for involving research participants (17893/001 & 002) and we were guided by a steering group of eight professionals.

**Provenance and peer review** Not commissioned; externally peer reviewed.

**Data availability statement** Data are available upon reasonable request. Please contact the corresponding author or unit manager (cpru.data@ucl.ac.uk) with enquiries about the data used in this study.

**ORCID iDs**
Claire Powell http://orcid.org/0000-0002-6581-0165
Gene Feder http://orcid.org/0000-0002-7890-3926
Ruth Gilbert http://orcid.org/0000-0001-9347-2709
Laura Paulauskaite http://orcid.org/0000-0002-7892-8103
Jenny Woodman http://orcid.org/0000-0002-9403-4177
Emma Howarth http://orcid.org/0000-0002-3969-7883

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
