## [Reviewer comments · BMJ Open]

ARTICLE DETAILS

TITLE (PROVISIONAL)	Child and family-focused interventions for child maltreatment and domestic abuse: Development of core outcome sets
AUTHORS	Powell, Claire; Feder, Gene; Gilbert, Ruth; Paulauskaite, Laura; Szilassy, Eszter; Woodman, Jenny; Howarth, Emma

VERSION 1 – REVIEW

REVIEWER	Krugman, Richard University of Colorado School of Medicine, The Kempe Center for the Prevention & Treatment of Child Abuse & Neglect
REVIEW RETURNED	23-May-2022

GENERAL COMMENTS	This is a comprehensive paper describing an enormous undertaking to attempt to identify the effectiveness of the myriad of programs that have been developed and studied that try to prevent and/or treat child abuse and neglect and Domestic Violence. The amount of material in this paper is overwhelming in its scope. The process is well described. Most impressive is the inclusion of "survivors" to be part of the process is a strength of the process and the paper. There are a few things that would be helpful and improve the discussion 1) It would be useful to know whether any programs have actually been evaluated using these COS. Have they been beta tested with actual service programs? 2) While the benefits to programs using this approach seem clear, what are the potential impediments to their use? Staffing? Training? Cost? In short, some link to how this work would be used in and by the C
--

REVIEWER	Victor, Bryan Wayne State University
REVIEW RETURNED	20-Jun-2022

GENERAL COMMENTS	Thank you for the opportunity to review this manuscript. The authors have undertaken an important and time-intensive study that will help to move the child maltreatment and domestic violence research fields forward through the establishment of two core outcome sets. A noted strength of the study is the inclusion of practitioners and survivors of child maltreatment and domestic violence within the process. Below I present one higher level concern and a few minor suggestions that I hope will help to strengthen the paper: On p. 3 authors note that “even widely reported outcomes such as depression or experience of violence are measured in varied ways” and frame this as problematic They also write that the “aim
---

	of a COS is to increase the consistency of outcome measurement". While I think this study moves the field forward by drawing attention to particular outcome domains, I don't think it resolves the issues raised on p. 3 of the introduction. As one example, authors put forward "child emotional health & wellbeing" as a core outcome for both CM and DVA. However, the representative constructs they suggest might be measured include: "emotions, mood, internalising problems, emotional security, and emotional numbness". This seems to open the door to a large variety of measurement strategies that would continue to generate the problems noted in the introduction. I think this issue merits further attention in the discussion section with recommendations or suggested steps for continuing to move evaluation research toward consistency of outcome measurement within the boundaries set up by the core outcome sets presented here. Minor suggestions The term "survivor" should be further defined in the abstract to clarify that this refers to survivors of child maltreatment and domestic violence. Would be helpful to provide a definition of DVA p. 6: Would be helpful to provide an example of how specific outcome indicators were combined together. pp. 7-8 Would be helpful to say a bit more about the concerns expressed by practitioners and researchers related to the size of the COS, and why the authors selected five as the appropriate number.
--	---

VERSION 1 – AUTHOR RESPONSE

Reviewer: 1

Dr. Richard Krugman, University of Colorado School of Medicine Comments to the Author: This is a comprehensive paper describing an enormous undertaking to attempt to identify the effectiveness of the myriad of programs that have been developed and studied that try to prevent and/or treat child abuse and neglect and Domestic Violence. The amount of material in this paper is overwhelming in its scope. The process is well described. Most impressive is the inclusion of "survivors" to be part of the process is a strength of the process and the paper. There are a few things that would be helpful and improve the discussion

1. *It would be useful to know whether any programs have actually been evaluated using these COS. Have they been beta tested with actual service programs?*

Programs have not yet been evaluated using the two COSs because another similar consensus process needs to take place to decide how best to measure the core outcomes. We have provided more detail about this in the highlighted sections in the discussion (lines 345-359) and the limitations (lines 385-394).

- 2) *While the benefits to programs using this approach seem clear, what are the potential impediments to their use? Staffing? Training? Cost?*

This answer to this question will depend very much on how the outcomes would be measured and whether the measurement tools require additional training or have proprietary costs associated with their use. It would also depend on how different these core outcomes are from those already

measured by services. From our current informal discussions with service providers, some but not all outcomes are currently measured by many services. We have provided more detail about this in the highlighted sections in the discussion (lines 345-359) and the limitations (lines 385-394).

3. *In short, some link to how this work would be used in and by the Child Protection System.* We have added a couple of sentences to the end of the limitations section to suggest how this could be used in the UK CPS: 'However, UK-based family and children's services already collect outcomes to support service planning and delivery, [30–32] so our hope is the CM- and DVA-COS would supplement pre-existing frameworks and support their harmonisation.'

Reviewer: 2

Dr. Bryan Victor, Wayne State University Comments to the Author:

Thank you for the opportunity to review this manuscript. The authors have undertaken an important and time-intensive study that will help to move the child maltreatment and domestic violence research fields forward through the establishment of two core outcome sets. A noted strength of the study is the inclusion of practitioners and survivors of child maltreatment and domestic violence within the process. Below I present one higher level concern and a few minor suggestions that I hope will help to strengthen the paper:

4. *On p. 3 authors note that "even widely reported outcomes such as depression or experience of violence are measured in varied ways" and frame this as problematic. They also write that the "aim of a COS is to increase the consistency of outcome measurement". While I think this study moves the field forward by drawing attention to particular outcome domains, I don't think it resolves the issues raised on p. 3 of the introduction. As one example, authors put forward "child emotional health & wellbeing" as a core outcome for both CM and DVA. However, the representative constructs they suggest might be measured include: "emotions, mood, internalising problems, emotional security, and emotional numbness". This seems to open the door to a large variety of measurement strategies that would continue to generate the problems noted in the introduction. I think this issue merits further attention in the discussion section with recommendations or suggested steps for continuing to move evaluation research toward consistency of outcome measurement within the boundaries set up by the core outcome sets presented here.*

Thank you for highlighting this. We do mention this in the limitations section and have added a couple of sentences in the discussion to emphasise this: 'In the shorter term, our core outcome sets can be used as a framework for service evaluation and research. Even if different measures are used across service evaluations, narrative methods can be used to synthesise findings across studies within a core outcome.'

Minor suggestions

5. *The term "survivor" should be further defined in the abstract to clarify that this refers to survivors of child maltreatment and domestic violence.*

We have added this to the participants section of the abstract: 'Stakeholders included CM and DVA survivors'

6. *Would be helpful to provide a definition of DVA*

We have added this to the Aim section: 'For DVA, we focused on interpersonal violence and abuse between parents/caregivers'.

7. *p. 6: Would be helpful to provide an example of how specific outcome indicators were combined together.*

We have added the following detail to the Stage 1: 'Longlist and taxonomy development section of the methods: 'For example, the child health outcome 'sleep' was created by combining: 'amount of sleep', 'quality of sleep', 'experience of nightmares', 'sleep routine', 'insomnia' and 'sleep-walking'.'

8. *pp. 7-8 Would be helpful to say a bit more about the concerns expressed by practitioners and researchers related to the size of the COS, and why the authors selected five as the appropriate number.*

We have provided the following detail in Stage 2 of the Results section: ‘Practitioners and researchers were concerned about the possible burden if the outcomes in the final COS were very different from those already collected. Through informal discussions with collaborators, five was agreed as enough to capture and compare shared outcomes but a small enough number that further outcomes could be added in research and evaluation without overwhelming service users.’

VERSION 2 – REVIEW

REVIEWER	Krugman, Richard University of Colorado School of Medicine, The Kempe Center for the Prevention & Treatment of Child Abuse & Neglect
REVIEW RETURNED	19-Aug-2022

GENERAL COMMENTS	The authors have addressed my concerns in the previous review. Best wishes with this important work.
--

REVIEWER	Victor, Bryan Wayne State University
REVIEW RETURNED	17-Aug-2022

GENERAL COMMENTS	The authors have thoroughly addressed each of the suggestions from my original review
---